# Effect of Impurity Adsorption on the Electronic and Transport Properties of Graphene Nanogaps

**DOI:** 10.3390/ma15020500

**Published:** 2022-01-10

**Authors:** Pablo Álvarez-Rodríguez, Víctor Manuel García-Suárez

**Affiliations:** 1Departamento de Física, Universidad de Oviedo, 33007 Oviedo, Spain; uo245071@uniovi.es; 2Centro de Investigación en Nanomateriales y Nanotecnología (CINN), 33007 Oviedo, Spain

**Keywords:** graphene nanogaps, density functional theory, tight-binding, quantum transport

## Abstract

Graphene stands out as a versatile material with several uses in fields that range from electronics to biology. In particular, graphene has been proposed as an electrode in molecular electronics devices that are expected to be more stable and reproducible than typical ones based on metallic electrodes. In this work, we study by means of first principles, simulations and a tight-binding model the electronic and transport properties of graphene nanogaps with straight edges and different passivating atoms: Hydrogen or elements of the second row of the periodic table (boron, carbon, nitrogen, oxygen, and fluoride). We use the tight-binding model to reproduce the main ab-initio results and elucidate the physics behind the transport properties. We observe clear patterns that emerge in the conductance and the current as one moves from boron to fluoride. In particular, we find that the conductance decreases and the tunneling decaying factor increases from the former to the latter. We explain these trends in terms of the size of the atom and its onsite energy. We also find a similar pattern for the current, which is ohmic and smooth in general. However, when the size of the simulation cell is the smallest one along the direction perpendicular to the transport direction, we obtain highly non-linear behavior with negative differential resistance. This interesting and surprising behavior can be explained by taking into account the presence of Fano resonances and other interference effects, which emerge due to couplings to side atoms at the edges and other couplings across the gap. Such features enter the bias window as the bias increases and strongly affect the current, giving rise to the non-linear evolution. As a whole, these results can be used as a template to understand the transport properties of straight graphene nanogaps and similar systems and distinguish the presence of different elements in the junction.

## 1. Introduction

The presence of strong and directional covalent bonds in graphene makes this material especially reliable and suitable for many applications, including biological sensors [1], organic electronics [2], or optoelectronics systems [3], to name a few. In particular, graphene has been proposed as an electrode in nanoscale molecular electronic systems, where molecules and other organic systems are used as bridges between separated graphene layers [4,5]. These new designs are paving the way for the fabrication of new components with robust and reproducible transport properties, as opposed to typical gold-based molecular electronics elements, where the lack of control on the coupling configuration on the gold surface translates into a lot of noise and a lack of reproducibility [6]. Graphene offers, on one hand, a stable layer whose atoms do not move nor migrate, and, on the other hand, a conducting surface on top of which nanoscale elements with large-area coupling groups [7,8,9,10] can sit with relatively high and stable binding energies. Such systems, which show universal features in the transmission across the molecule [11], are expected to increase the reliability of the transport properties and lead to faster, more stable and scalable electronic components.

The first step towards the design of graphene-based molecular electronic systems is the fabrication of a nanogap, i.e., a physical separation between two graphene layers of the order of nanometers or even smaller. This has already been achieved by some groups worldwide, which were able to create such gaps by using electroburning techniques [5,12,13,14] or mechanically controllable break junctions [15]. The transport properties across these gaps were also measured and it was found that in some cases, reproducible transport characteristics appear that signal the presence of wedges facing straight edges [16]. The occurrence of wedges and other impurities can lead to the development of a plethora of electronic functionalities (rectification, negative differential resistance, magnetoresistance, and spin filtering). These functionalities and the stability associated to the edges would make it possible for the design of electronic components to have just the electrodes separated by a nanogap, making it unnecessary for the presence of bridging elements [17].

To properly characterize the electronic and transport properties of a nanogap it is necessary to determine the types of impurities that attach to the carbon atoms at the edges and passivate them. This is currently something difficult to achieve in an experiment, since the fabrication of nanogaps is still in its infancy. In particular, it is still not possible the control over the composition and morphology of the gaps, making it rather difficult to regulate and characterize the presence of impurities that attach to the graphene edges once the gap is created. The cleanliness of the gap can in principle be improved by using ultra-high vacuum chambers or certain atmospheres, but no certainty of the specific impurities that attach to different parts of the edges can be obtained.

We show here that the transport characteristics of graphene nanogaps with straight edges can indeed be used to determine the type of impurities that attach to a given edge. Such characteristics depend strongly on the type of atom that passivates the edge and can be taken as a starting point to fully characterize its composition and configuration, opening the door to the development of nanogap-based electronic components [16,17]. We use a series of atoms from the first and second rows of the periodic table (H, B, C, N, O, and F), which are common impurities in the experiments and whose electronic properties and their evolution from one to another are well known. We attach these atoms to straight graphene electrodes with an armchair edge. We find that the transport properties (transmission and current) strongly depend on the type of atom that attaches to the edge. The evolution of such properties with the type of atom can also be easily explained by using a simple tight-binding model. We show then that by analyzing the transport properties of graphene edges with different impurities attached to them, it is possible to unambiguously determine their type of passivation.

The article is structured as follows. In the next section, we explain the methods we used to calculate the electronic and transport properties of graphene nanogaps: Ab-initio simulations and tight-binding model. Next we present the main results and discuss them. We finish with the conclusions.

## 2. Methods

The electronic structure was calculated with density functional theory (DFT) [18], as implemented in the Siesta code [19]. Siesta uses norm-conserving pseudopotentials [20] to get rid of the core electrons and linear combinations of atomic orbitals (LCAO) to span the valence states. A double-ζ polarized (DZP) basis set defined with an energy shift of 0.001 Ry. The real space grid was defined with a 200 Ry cutoff. Exchange and correlation were calculated with the local density approximation (LDA) and parameterized with the Ceperly–Alder functional [21]. The graphene layer was initially simulated with the smallest possible unit cell along the *y* direction. Increasing the size of the unit cell or the number of *k*-points along *y* gave essentially the same results around the Fermi level, but dramatically changed the current in some cases, as we shall see. A vacuum region of 20 Å along the *x* direction was included to make sure the periodic images did not overlap along this direction. A total of 12 atoms and 10 *k*-points along the *z* direction were used in the simulation of the electrodes and 62 atoms were employed in the simulation of the scattering region (for 1 unit cell along *y*).

The transport calculations were carried out using the Gollum code [22], which gets the electronic structure from the Siesta calculation: The Hamiltonian of the bulk electrodes was calculated in a separate simulation, which included, as stated before, *k*-points along the transport direction to reproduce the bulk electronic structure. The Hamiltonian of the scattering region was computed in a simulation that included the gap, edges, and some bulk unit cells towards positive and negative *z* values to make sure the electronic structure converged to that of the bulk away from the gap. The effect of semi-infinite electrodes was then introduced as self-energy matrices that were computed from the electrodes Green functions. This assumed then that the electrodes are ideal semi-infinite graphene layers without additional contacts; notice, however, that the use of more realistic electrodes, which would include contacts between the graphene layers and metallic electrodes, could decrease the transmission and conductance [23]. The zero-bias transmission was calculated from the imaginary part of the self-energies and the retarded Green function of the scattering region, and the zero-bias conductance was calculated as the transmission at the Fermi level. The temperature-dependent conductance, calculated as $G\left(T\right)=\int_{-\infty }^{\infty }T\left(E\right)\left(-\partial f\left(E,T\right)/\partial E\right)dE$, gives essentially the same value for a large range of temperatures (well above room temperature), since the zero-bias transmission around the Fermi level is smooth and featureless. When a bias was implemented, the finite-bias transmission and current were calculated directly from the equilibrium electronic structure by shifting the levels of both sides of the extended molecule [24]. This methodology was specially justified in this case, since the potential in the scattering region declines almost entirely in the gap, where there are no states.

In order to physically interpret the results and understand the trends behind the ab-initio transport results, a one-dimensional tight-binding model that can properly reproduce the transmission and current between graphene electrodes was also used [17]. This model, whose Hamiltonian is $\hat{H}={\hat{H}}_{l}+{\hat{H}}_{r}+{\hat{V}}_{\mathrm{lr}}$, considers on each side of the gap the interaction between carbon bulk states, or states which belong to other atoms that couple strongly to the bulk continuum, i.e.,: (1)$$\begin{array}{cc} {\hat{H}}_{l(r)} & =\sum_{\langle i,j\rangle ;\sigma }(t_{ij\sigma }\overset{+}{\left(-\right)}\delta_{ij}eV/2){\hat{c}}_{i\sigma }^{†}{\hat{c}}_{j\sigma }+\\ & +\sum_{\sigma }\left(\epsilon_{\sigma }\overset{+}{\left(-\right)}eV/2\right.{\hat{d}}_{\sigma }^{†}{\hat{d}}_{\sigma }+\sum_{\sigma }t_{l(r),1d\sigma }\left({\hat{c}}_{1\sigma }^{†}{\hat{d}}_{\sigma }+h.c.\right) \end{array}$$
where the fermion operators $\left\{{\hat{c}}^{†}\right\}$ and $\left\{{\hat{d}}^{†}\right\}$ create electrons in the inner/bulk sites and in the other sites at the edge, respectively, and σ=↑,↓ represents the spin. We use in the bulk $t_{ij}=-3$ eV for i≠j and $t_{ii}=0$ eV (the same value for both spins). The coupling between the bulk states and edge state, $t_{l(r),1d\sigma }$, is very small in this case [17] and only affects when interference effects are present (see below). The on-site $\epsilon_{\sigma }$ depends also on the type of impurity. The bias voltage was included as a rigid shift of the left and right states. In this case, this is again a valid approximation that coincides with more involved non-equilibrium Green’s functions (NEGF) calculations [24] because there is no element in the middle and all the potential falls in the gap.

The absolute value of the transmission at the Fermi level depends on the strength of the ${\hat{V}}_{lr}$ term and its slope is determined by the on-site energies $\epsilon =t_{11}$ of the carbon atoms previous to the edge. If ϵ is higher (lower) than the on-site energy of the bulk carbon atoms then the slope is positive (negative) due to the increase of the transmission at the position of the on-site energy and its reduction away from it. In addition, in some cases, due to the finite width of the ribbons in the ab-initio simulations and the presence of various atoms along the perpendicular *y* direction, the atoms at the edge and near it can affect the transmission by inducing interference effects. These effects can also be simulated with the model by including additional couplings between the carbon and edge atoms [17], i.e.,
(2)$${\hat{V}}_{\mathrm{lr}}=\sum_{\sigma }\left[\gamma_{\mathrm{dd}\sigma }{\hat{d}}_{l,\sigma }^{†}{\hat{d}}_{r,\sigma }+\sum_{i,j=1}^{2}\left(\gamma_{di\sigma }{\hat{d}}_{l,\sigma }^{†}{\hat{c}}_{r,i\sigma }+\gamma_{id\sigma }{\hat{c}}_{l,i\sigma }^{†}{\hat{d}}_{r,\sigma }+\gamma_{ij\sigma }{\hat{c}}_{l,i\sigma }^{†}{\hat{c}}_{r,j\sigma }\right)+h.c.\right]$$
where the $\gamma_{\alpha \beta \sigma }$ are the coupling elements between a state α on the left and a state β on the right for spin σ.

## 3. Results and Discussion

We consider armchair edges, which means the direction along which the electrodes are grown, and is the perpendicular transport direction (*z*), to be zigzag. The structure can be seen in Figure 1. The armchair edges are non-magnetic when passivated with hydrogen, which implies the density of states is the same for spin up and down, as shown in Appendix A. We show the projected density of states (PDOS) on the edge atoms for one side of the gap without coupling to the other side (the weak coupling across the gap in most cases slightly modifies this picture). For other elements, however, there can be some magnetic signal, as in the cases of boron, nitrogen, and oxygen. The magnetic moment is shown in Table 1. As can be seen, the magnetic moment in case of hydrogen passivation is zero, a result which is well known for this type of edge. However, for other elements such as boron, nitrogen, and, specially, oxygen, the magnetic moment has a finite value. These finite values can translate into a magnetic signal in the transport properties at low temperatures, as we shall see. By closely inspecting the Mulliken populations it can also be found that the magnetic moment in these cases comes mainly from the atoms at the edge, while those carbons connected to them do not develop any sizeable value.

The PDOS presented in Appendix A also show the presence of some sharp peaks that move to lower energies from boron to fluorine. Such peaks come from the p${}_{x}$ and p${}_{y}$ orbitals, which do not overlap strongly with the orbitals of the carbon atoms of graphene and give then rise to sharp features. The p${}_{z}$ and s orbitals do however overlap with the corresponding orbitals of the carbons and therefore spread over the shown energy range. The movement to lower energies from boron to carbon is the expected behavior, as one can expect from the increase of the nuclear attraction. On the other hand, the PDOS of the carbon atoms in contact with the atoms at the edge, also shown in Appendix A, are very similar for all atoms but fluorine, which present a larger hybridization that indicates a much stronger coupling. In addition, the PDOS of the carbons do not show any sizeable spin split in case of H, B, and C, shown in panels (a), (b), and (c) of Figure A2, which indicates that the magnetic moment of the magnetic atoms at the border does not influence the neighboring atoms of the graphene layer. In case of N, O, and F, shown in panels (d), (e), and (f) of Figure A2, there is however a spin splitting around the Fermi level, which indicates that in these cases the carbon atoms acquire some magnetic moment.

Once the electronic and magnetic properties are calculated, we simulate the electronic transport. Notice first that the bulk states, whose local density of states (LDOS) integrated around the Fermi level is shown in Figure 1, reach the carbon atoms at edges, which implies their coupling across the gap is going to be larger than for zigzag edges, which do not couple to the edge states [16,17]. We first determine the zero-bias transmission at a distance between the edges (size of the nanogap) of 3 Å. A compilation of the results is shown in Figure 2, where we plot the transmission across graphene nanogaps passivated with hydrogen, which is the most typical case, and the other atoms from the second row of the periodic table (B, C, N, O, and F). Notice that, as one moves from B to F, two trends can be observed: First, the transmission at the Fermi level (conductance) steadily decreases and, second, its positive slope is also reduced. This evolution can be easily explained by taking into account two effects: On the one hand, as one moves towards the right of the periodic table the effective size of the atom decreases due to the increase of the nuclear attraction, which reduces the overlap across the gap; on the other hand, the on-site energies move to lower energies for the same reason as before, which reduces the slope of the transmission. The change in the slope can be properly simulated with the tight-binding model [17] and easily explained by taking into account that the maximum of the transmission is at the energy of the on-site state, which is initially above the Fermi level; as this moves to lower energies the maximum moves as well, decreasing the value of the slope and eventually changing its sign.

The transmission also changes sharply outside a window between −2.5 and 2.0 eV around the Fermi level. Below −2.5 eV, the transmission increases, while it decreases above 2.0 eV. These changes of the transmission can be explained by taking into account the presence of band-edges, due to the finite-width of the graphene layers, and the couplings between the edge atoms and graphene layer. From the PDOS, it can be seen that the p${}_{z}$ and s states of the edge atoms couple to those of the carbon atoms and also, but more weakly, to the perpendicular π states above the Fermi level. This translates into a higher transmission for the band with p${}_{z}$ states at the Fermi level and a lower transmission for the π states above the Fermi level. Below the Fermi level, the states of the carbon band at these energies couple also to the perpendicular states of the edge atoms, increasing thus the transmission.

The exponential decay of the conductance, obtained by calculating the transmission at the Fermi level as a function of the distance between the edge atoms on both sides of the gap, can be seen in Figure 3. A clear exponential behavior can be observed in most cases, which is typical of the tunneling regime. The β factors, obtained by fitting the curves, are shown in Table 1. Note that these are approximate values, since some of the curves deviate from a pure exponential behavior at the beginning or at the end (e.g., the H and B curves; the other cases fit rather well to an exponential). From these results, it can be deduced that the closer the atom to the right of the periodic table, the higher the β, which, again, is an expected result due to the weakening of the coupling and the movement of the on-site energies to lower values, which translates into a higher tunneling barrier.

The IV characteristics are shown in Figure 4. According to the tight-binding model, for straight and featureless edges without localized states [17], the IV curve should be ohmic. However, we find that its shape strongly depends on the size of the simulation cell along the perpendicular *y* direction. As can be seen, when one unit cell is used to repeat the nanoribbon along the perpendicular direction, the current is ohmic at low voltages (lower than 0.25 eV) but then develops a rather non-linear behavior, with peaks and negative differential resistances (NDR). This behavior is not modified when *k*-points are used along *y*. The NDR features are in general rather sharp, with acute increases and decreases to almost zero values, something desirable for devices such as oscillators and amplifiers. Notice, however, that these NDR effects, which are prominent in the case of a coherent and ballistic transport regime, like the one assumed here, could be reduced when dissipative effects are taken into account ([25] and references therein).

The origin of the non-linear behavior can be explained by taking into account the appearance of interference-like features that come from high energies and move towards the Fermi level, decreasing thus the transmission and current. Some of these features can be seen in Figure 5. These are typical interference traits (one peak followed by a dip or vice versa, a Fano resonance, or one or various dips, a multiple path feature) that introduce sharp features in this otherwise smooth transmission. These features can be reproduced with the tight-binding model by including different couplings between the “carbon” sites and the impurity site at the edge (which originally do not affect the transmission and current at small voltages because they do not couple to other states in the same side or across the gap). For instance, if a coupling between the impurity site and the next “carbon” site on the same side $t_{1d}$ is included, which produces a dip that appears in the transmission at the energy of the impurity site. If, however, a coupling between the impurity site and the second “carbon” site on the same side, $t_{2d}$, is included, this gives a peak and a dip, a typical Fano feature shown in Figure 5b (red dashed curve). If both of these couplings are included, they give a dip that transforms into a dip and a peak as the coupling with the second site increases. These features correspond to what we call localized-state coupling (LSC), because they represent situations where there is a coupling between the localized state given by the impurity and other states in the same side. As an example, consider, for instance, a particular orbital of the impurity atom in the ab-initio calculation with a certain on-site energy that might not couple for symmetry and spatial reasons to a given orbital in an nearest neighbor, but could couple to the same orbital in a second nearest neighbor; this particular coupling, conceptualized with the ab-initio model, can then give rise to this particular feature of the transmission at certain voltages.

Other features that appear at higher voltages in the ab-initio simulation and which come also from interference (various peaks and/or dips) can also be explained by taking into account couplings across the gap of the impurity sites (what we call second-order coupling, 2OC) [17]. For instance, if a coupling is introduced between the impurity in one side and the second “carbon” atom on the other side, $\gamma_{d2}$, this gives a dip, two peaks, and another dip, a multiple path features shown also in Figure 5b (green dot-dashed curve). This feature, as well as the LSC one, move towards the Fermi level as the bias increases and also change their shape due to the shift of the on-site energies, a behavior that is also captured by the tight-binding model. These two cases (LSC and 2OC) essentially characterize all features that might appear in the transmission as the voltage increases. Notice, however, that such cases represent a qualitative characterization of the essential features that appear in a more complex system (which has different on-site orbitals and couplings between them at different energies and distances) and serve to isolate such features and explain their origin (the particular coupling that becomes more relevant for a given energy and voltage.)

These interference features become then important in the case of just one graphene unit cell along *y* and are responsible for the strongly non-linear behavior of the current in such a case. When two unit cells are used along *y*, however, the current becomes ohmic for the full voltage interval, up to 1.5 V or even higher, which is also in agreement with the simple tight-binding model without additional couplings. The same exact result is found when the number of unit cells is increased to three along *y*. This shows that the interference-related features disappear when the number of atoms along the perpendicular directions increases and seems to indicate that the non-linear behavior observed for one unit cell along *y* is a finite-size effect that comes from a particular design of the system with just one unit cell. We do not therefore exclude that such non-linear effects appear when truly finite-width nanoribbons are used, something that would need further investigation beyond the scope of this article. We can then conclude that it is necessary to use at least two unit cells to properly simulate the transport properties of armchair broken graphene sheets or wide ribbons.

All ohmic currents calculated with two unit cells along *y* are compared in Figure 6, where it is clear, just like for the transmissions and conductances, that the current decreases as one moves towards the right of the periodic table. These ohmic currents can be used, first, to prove that the edges are straight and do not have prominent features, and, second, to distinguish the type of impurity that is attached to the edges. By carefully measuring then the current in present experiments and mapping such a current to atmospheres with certain types of impurities, it should be possible to establish a clear relation between both of them.

## 4. Conclusions

By means of ab-initio simulations and a tight-binding model, we showed that the electronic and transport properties of graphene nanogaps with straight armchair edges depend substantially on the type of passivation of the edges. In particular, we obtained that such properties strongly depend on the the size and on-site energy of the atom that is used for the passivation. The size of the atom decreases the transmission and gives rise to smaller conductances and currents, i.e., the smaller the size of the atom, the smaller the conductance, and current. The on-site energy changes the slope of the transmission at the Fermi level and also affects the exponential decay. The smaller the on-site energy, the smaller the slope and bigger the β value.

We also found that the IV characteristics strongly depend on the size of the system along the perpendicular direction. For small systems (one armchair unit cell along *y*), the current turns out to be highly non-linear due to the presence of interference effects (Fano resonances and other interference-related features) that enter into the bias window at high voltages. The origin of these resonances can be traced back, with the help of the ab-initio model, to couplings between the carbon closest to the edge and side states (Fano states) that come from atoms located along the perpendicular direction. Other couplings between atoms across the gap can also give rise to additional interference features. When the size along the perpendicular direction is increased, however, such interference effects disappear and the current becomes ohmic. These results are useful, on one hand, to univocally determine the passivation of graphene nanogaps in current molecular electronics experiments with graphene electrodes and pave the way, on the other hand, for future studies of finite-size and interference effects in the transport properties of these systems.

## Figures and Tables

**Figure 1 materials-15-00500-f001:**
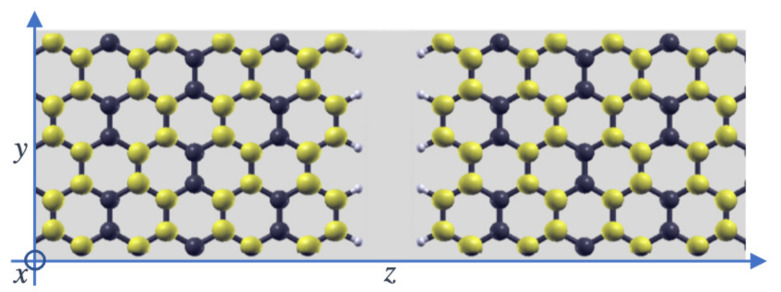
Graphene nanogap with armchair edges. The yellow lobes represent the local density of states (LDOS) integrated in a small energy window around the Fermi level.

**Figure 2 materials-15-00500-f002:**
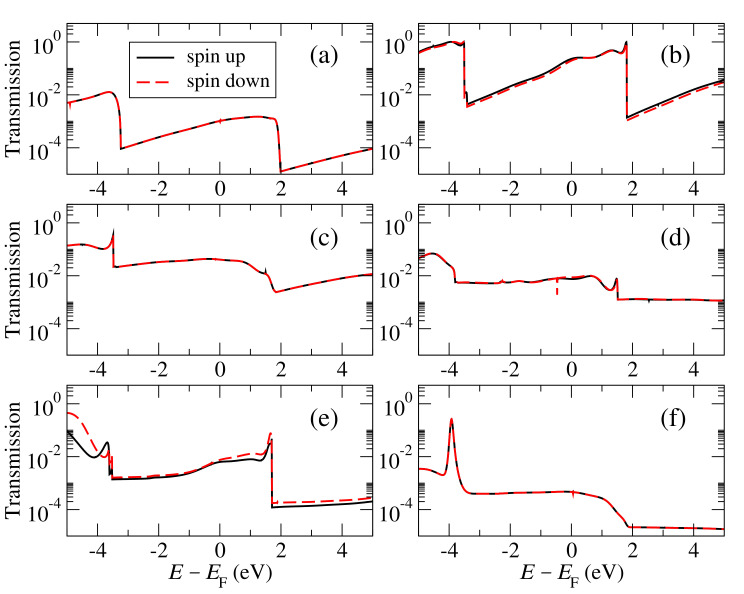
Transmission as a function of energy for graphene nanogaps with an armchair edge and passivated with H (**a**), B (**b**), C (**c**), N (**d**), O (**e**), and F (**f**).

**Figure 3 materials-15-00500-f003:**
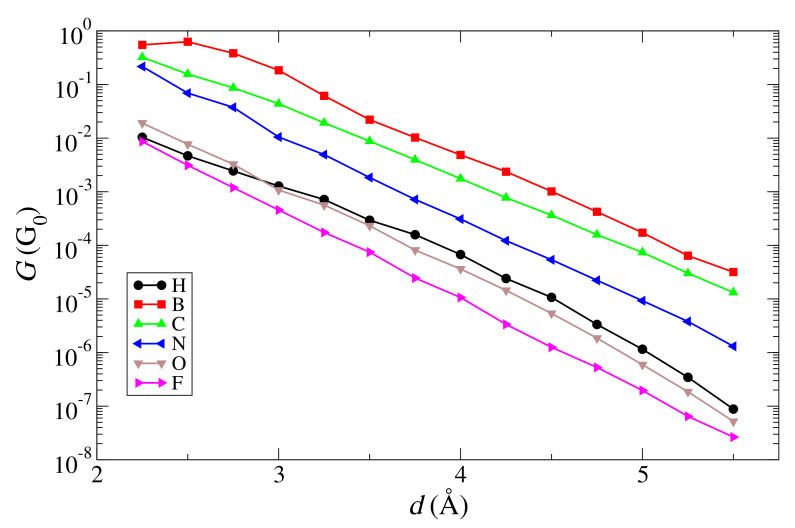
Conductance as a function of distance for graphene nanogaps with an armchair edge and passivated with H, B, C, N, O, and F.

**Figure 4 materials-15-00500-f004:**
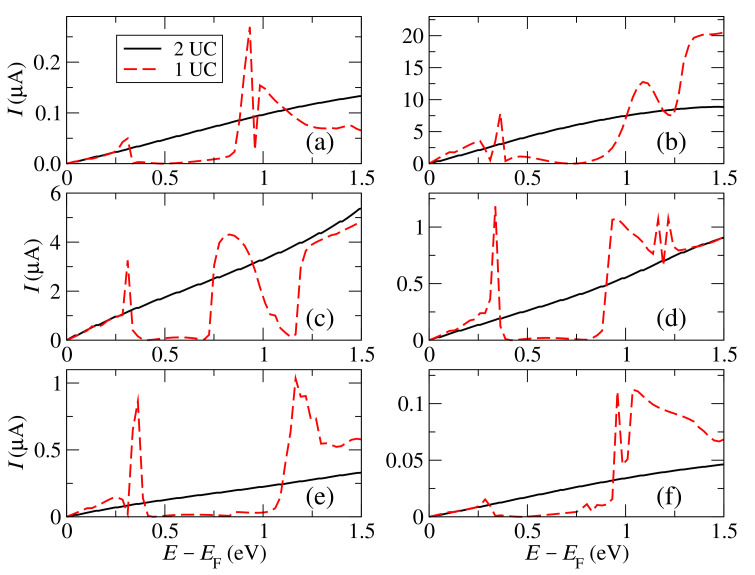
Total current versus voltage for graphene nanogaps with an armchair edge, one or two unit cells (UC) along the perpendicular direction (*y*) and passivated with H (**a**), B (**b**), C (**c**), N (**d**), O (**e**), and F (**f**).

**Figure 5 materials-15-00500-f005:**
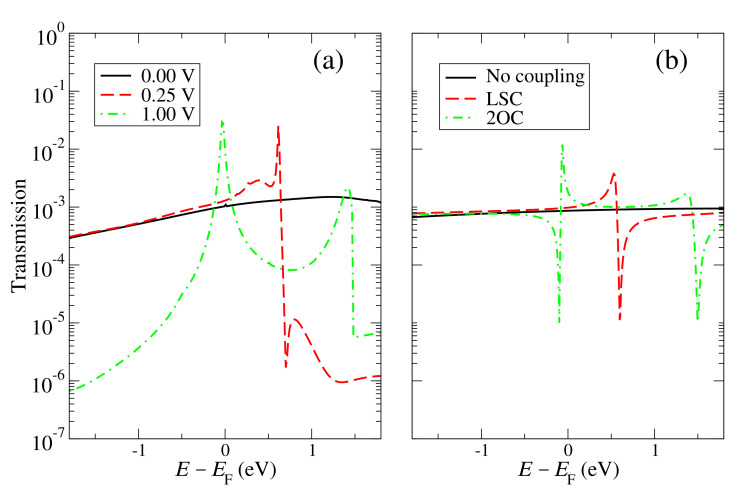
Transmission at 0.00 V and 0.25 V for the ab-initio calculations (**a**) and the tight-binding model without any additional coupling, with localized-state coupling (LSC) and with second-order coupling (2OC) (**b**). In the tight-binding model, the coupling across the gap between the carbon atoms closest to the edge is $\gamma_{11}=-0.045$ eV; in the LSC case, the impurity on-site energy is ϵ=1.0 eV, the coupling between the second closest carbon atom and the impurity in the same layer is $t_{2d}=-0.6$ eV and the bias voltage is 1.8 V; in the 2OC case, the impurity on-site energy is ϵ=0.2 eV, the coupling between the impurity and the inner carbon atoms on the other layer is $\gamma_{d2}=-0.6$ eV and the bias voltage is 1.6 V.

**Figure 6 materials-15-00500-f006:**
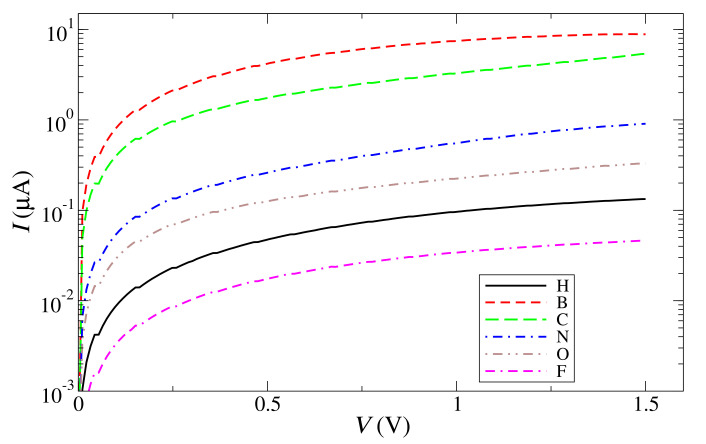
Total current versus voltage for graphene nanogaps with an armchair edge and passivated with H, B, C, N, O, and F.

**Table 1 materials-15-00500-t001:** Equilibrium distance (*d*) between the edge atom and the carbon atom, magnetic moment (*m*), total conductance (*G*) for both spins at a distance of 3 Å, and decaying factor (β) for graphene armchair nanogaps with edges passivated with different elements.

Element	Equilibrium Distance *d* (Å)	Magnetic Moment *m* ($\mu_{B}$)	Conductance *G* (G${}_{0}$)	Decaying Factor β (Å${}^{-1}$)
H	0.98	0.00	1.04 × 10${}^{-3}$	3.3
B	1.39	0.15	2.10 × 10${}^{-1}$	2.9
C	1.22	0.00	4.07 × 10${}^{-2}$	3.0
N	1.19	0.17	8.18 × 10${}^{-3}$	3.7
O	1.23	0.50	6.84 × 10${}^{-3}$	3.8
F	1.25	0.00	4.64 × 10${}^{-4}$	3.8

## Data Availability

The data supporting the reported results can be provided by email.

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
