# Peer review of "Effect of Impurity Adsorption on the Electronic and Transport Properties of Graphene Nanogaps"

_materials, 2022, doi:10.3390/ma15020500_

Round 1

Reviewer 1 Report

The Authors reported the theoretical investigation of graphene nanogaps by using ab initio electronic and transport calculations. They studied the effect of different impurity atoms in nanogaps and observed correlation between the transport values and the size of the atoms. Although the topic of the impurities in nanogaps can be interesting for the field of nanoelectronics, unfortunately both the methodology and the interpretation of the results are not sufficient for publication in a scientific journal. Some of my comments are listed as follows:

1, The tight-binding methodology has been poorly described. The coupling terms are not interpreted clearly in the text and the Authors do not give any kind of TB parameters; therefore, the results are not reproducible.

2, One of the main findings in the paper is related to the highly non-linear behavior with negative differential resistance. However, after the presentation of these results, the Authors explicitly write that the effect is only a computational artefact (see below):

line 325: “seems to indicate that the non-linear behavior observed for one unit cell along y is a finite-size effect, which can be considered an artifact of the calculation in broken graphene sheets”.

3, The Authors claim that the carbon atoms next to the impurity atoms do not have any sizable spin split (line 157). However, Figure A2 “e” and “d” clearly show splitted spin up and spin down PDOS around the Fermi-level.

3, Unfortunately, it is difficult to understand the interpretations of the numerical results. Just one example (from several ones), where the sentence does not have clear meaning for the Reviewer:

“These features can be also reproduced with the tight-binding model by including the coupling between those atoms at the edge or close to it and side states; in particular, the coupling between the atoms just before those of the border (carbon atoms) and a localized state (Fano state), what we call localized-state coupling, LSC.”

4, The present version of the paper does not even contain a reference list. (at least the version what the Reviewer received)

Reviewer 2 Report

The authors used ab-initio simulations and a tight-binding model to investigate the electronic and transport properties of graphene nanogaps with straight armchair edges passivated with different atoms (H, B, C, N, etc.). They showed that the properties of these nanogaps strongly depend on the type of passivation, i.e. on the size and on-site energy of the atom that is used for the passivation. It was reported that increasing the size of the atoms decreases the transmission, and consequently the conductance and current through the nanogap. Similarly, smaller on-site energy results in smaller slope of the transmission at the Fermi level, and larger value of the exponential decay parameter. Finally, the authors demonstrate simple ohmic behavior in I-V characteristic for wider nanoribbon contact regions, whereas a non-linear behavior is reported for the narrowest device due to quantum interference effects. The motivation is described clearly in the introduction, methodology is appropriate, and the manuscript is interesting and mostly well-written, however, there are several issues that need to be resolved before I can recommend it for publication.

Major:

(1) As the authors have said, the issue of using proper contacts is crucial in molecular electronic devices. The authors use an approximation for electrodes, i.e. assuming they are ideal semi-infinite nanoribbons identical to edge parts of the scattering region. This is a usual and widely-used approximation that results in perfect unitary transmission for each conducting mode. However, realistic electrodes would introduce broadening that completely changes the electronic and transport properties of the nanostructure, see e.g. DOI: 10.3390/ma14133670. The authors should mention and shortly discuss these limitations in their work.

(2) The authors state that "zero-bias conductance was calculated as the transmission at the Fermi level". This means that conductance is calculated at 0K? Why not use the finite-temperature Fermi function to find the conductance at e.g. room temperature, which would be more relevant for electron device applications?

(3) Non-linear I-V characteristics, including negative differential resistance (NDR) regions, are expected in low-dimensional devices such as these, especially if the transport is coherent and ballistic, as in this study. If the authors employed dissipative quantum transport simulations, most likely all NDR effects would be reduced or would disappear completely, see e.g. papers on NEGF simulation of resonant tunneling devices by Gerhard Klimeck group at Purdue. Therefore, I would not put much importance in results concerning NDR.

(4) Why are the differences between DFT and tight-binding models in Fig. 5 so significant? One would expect better match for a single-purpose TB model. Also, it is unclear for which voltage Fig.5b is obtained?

Minor:

(5) Please insert axes in Fig. 1 so the reader can know which direction is x, y, z.

(6) Please improve the readability of Fig. 6. When printed b/w, lines are not discernible. I suggest changing line styles, or attaching letters (H, B, C, etc.) to each of the lines.

(7) There are typos ("sencond", "lineal", etc.) at several places in the text, please do a spell-check during revision.

Round 2

Reviewer 1 Report

The authors have satisfactorily answered to my comments and I found that the manuscript has been improved significantly. Consequently, I recommend this work for publication in MDPI Materials.

Reviewer 2 Report

The authors have done a thorough revision and the manuscript is now in much better shape. All comments have been properly addressed in the revised version. I recommend this paper for publication.